# CONGESTED BANDITS:
# OPTIMAL ROUTING VIA SHORT-TERM RESETS

## ABSTRACT

For traffic routing platforms, the choice of which route to recommend to a user depends on the congestion on these routes – indeed, an individual's utility depends on the number of people using the recommended route at that instance. Motivated by this, we introduce the problem of Congested Bandits where each arm's reward is allowed to depend on the number of times it was played in the past $\Delta$ timesteps. This dependence on past history of actions leads to a dynamical system where an algorithm's present choices also affect its future pay-offs, and requires an algorithm to plan for this. We study the congestion aware formulation in the multi-armed bandit (MAB) setup and in the contextual bandit setup with linear rewards. For the multi-armed setup, we propose a UCB style algorithm and show that its policy regret scales as $\tilde{O}(\sqrt{K\Delta T})$. For the linear contextual bandit setup, our algorithm, based on an iterative least squares planner, achieves policy regret $\tilde{O}(\sqrt{dT} + \Delta)$. From an experimental standpoint, we corroborate the no-regret properties of our algorithms via a simulation study.

## 1 INTRODUCTION

The online multi-armed bandit (MAB) problem and its extensions have been widely studied and used to model many real world scenarios (Robbins, 1952; Auer et al., 2002; 2001). In the basic MAB setup there are $K$ arms with either stochastic or adversarially chosen reward profiles. The goal is to design an algorithm that achieves a cumulative reward that is as good as that of the best arm in hindsight. This is quantified in terms of the regret achieved by the algorithm over $T$ time steps (see Section 2 for formal definitions). In many real world scenarios the MAB setup as described above is not suitable as the reward obtained by playing an arm/action at a given time step may depend on the algorithm's choices in the previous time steps. In particular, we are motivated by online routing problems where the reward of suggesting a particular edge to traverse along a path from source to destination often depends on the congestion on that edge. This congestion is a function of the number of times the particular edge has been recommended earlier (potentially in a time window). In such scenarios, one would desire algorithms that can compete with the best *policy*, i.e., the best sequence of actions, in hindsight as compared to a fixed best action.

Classical multi-armed bandit formulation and associated no-regret algorithms (Slivkins, 2019), or their extensions to routing problems (Kalai & Vempala, 2005; Awerbuch & Kleinberg, 2008) do not suffice for the above scenario as they only guarantee competitiveness with respect to the best fixed arm in hindsight. To overcome these limitations, we propose a new model, *viz.*, *congested bandits* which captures the above scenario. In our proposed model the reward of a given arm at each time step depends on how many times the arm has been played within a given time window of size $\Delta$. Hence over time, an arm's expected reward may decay and reset dynamically. While our model is motivated by online routing problems, our proposed formulation is very general. As another example, consider a digital music platform that recommends artists to its end users. In order to maximize profit the recommendation algorithm may prefer to recommend popular artists, and at the same time the platform may want to promote equity and diversity by highlighting new and emerging artists as well. This scenario can be model via congested bandits where each artist is an arm and the reward for suggesting an artist is a function of how many times the artist has been recommended in the past time window of length $\Delta$. In both the scenarios above, the ability to reset the congestion cost (by simply not playing an arm for $\Delta$ time steps) is a crucial part of the problem formulation.

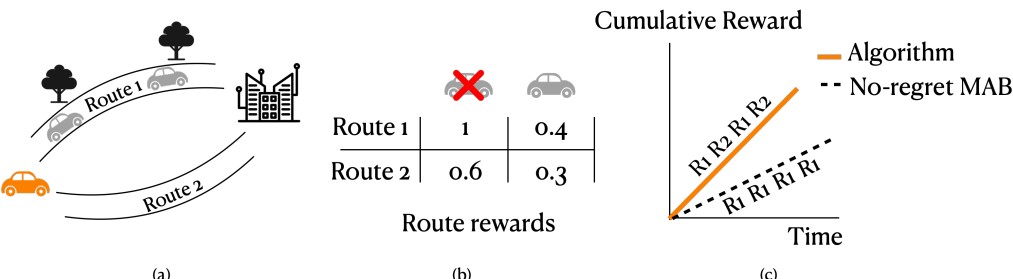

**Figure 1.** Our proposed Congested Bandits framework. (a) A route recommendation scenario where an algorithm can recommend one of two routes to the incoming vehicle. (b) The reward for each route depends on whether there is congestion on the route or not. (c) Traditional multi-armed bandit algorithms learn to recommend the best route, Route 1 for every incoming vehicle. This is clearly suboptimal. Our algorithm, CARMAB, adapts to the congestion and achieves better performance.

**Our contributions.** We propose and study the *congested bandits* model with *short term resets* under a variety of settings and design no-regret algorithms. In the most basic setup we consider a $K$-armed stochastic bandit problem where each arm $a$ has a mean reward $\mu_a$, and the mean reward obtained by playing arm $a$ at time $t$ equals $f_{\text{cong}}(a, h_t)\mu_a$. Here $h_t$ denotes the history of the algorithm's choices within the last $\Delta$ time steps and $f_{\text{cong}}$ is a non-increasing congestion function. Recall that the algorithm's goal in this setup is to compete with the best policy in hindsight. While the above setting can be formulated as a Markov Decision Process (MDP), existing no regret algorithms for MDPs will incur a regret bound that scales exponential in the parameters (scaling as $K^{\Delta}$) (Auer et al., 2009; Jaksch et al., 2010). Instead, we carefully exploit the problem structure to design an algorithm with near-optimal regret scaling as $\tilde{O}(\sqrt{K\Delta T})$. Next, we extend our model to the case of online routing with congested edge rewards, again presenting near optimal no-regret algorithms that avoid exponential dependence on the size of the graph in the regret bounds.

We then extend the multi-armed and the congested online routing formulations to a contextual setting where the mean reward for each arm/edge at a given time step equals $f_{\text{cong}}(a, h_t)\langle\theta_*, \phi(x_t, a)\rangle$, where $x_t \in \mathbb{R}^d$ is a context vector and $\theta_*$ is an unknown parameter. This extension is inspired from classical work in contextual bandits (Li et al., 2010; Chu et al., 2011) and captures scenarios where users may have different preferences over actions/arms (e.g., a user who wants to avoid routes with tolls). Solving the contextual case poses significant hurdles as a priori it is not even clear whether the setting can be captured via an MDP. By exploiting the structure of the problem, we present a novel epoch based algorithm that, at each time step, plays a near optimal policy by planning for the next epoch. Showing that such planning can be done when only given access to the distribution of the contexts is a key technical step in establishing the correctness of the algorithm. As a result, we obtain algorithms that achieve $\tilde{O}(\sqrt{dT} + \Delta)$ regret in the contextual MAB setting. Finally, using simulations, we perform an empirical evaluation of the effectiveness of our proposed algorithms.

**Related work.** Closest to our work are studies on multi-armed bandits with decaying and/or improving costs. The work of (Levine et al., 2017) proposes the rotting bandits model that has been further studied in (Seznec et al., 2019). In this model the mean reward of each arm decays in a monotonic way as a function of the number of times the arm has been played in the past. There is no notion of a short-term reset as in our setting. As a result it can be shown that a simply greedy policy is optimal in hindsight. In contrast, in our setting greedy approaches can fail miserably as highlighted in Figure 1. The work of Heidari et al. (2016) considers a setting where the mean reward of an arm can either improve or decay as a function of the number of times it has been played. However, similar to the rotting bandits setup there is no notion of a reset. Pike-Burke & Grünewälder (2019) considers a notion of reset/improvement by allowing the mean reward to depend on the number of time steps since an arm was last played. Their work considers a Bayesian setup where the mean reward function is modelled by a draw from a Gaussian process. On the other hand, in our work, the reward is a function of the number of times an arm was played in the past $\Delta$ time steps. However their regret bounds are either with respect to the instantaneous regret, or with respect to a weaker policy class of $d$-step look ahead policies. The notion of instantaneous regret only compares with the class of greedy policies at each timestep as compared to the globally optimal policy.

There also exist works studying the design of online learning algorithms against $m$-*bounded memory* adversaries. The assumption here is that the reward of an action at each time step depends only on the previous $m$ actions. While this is also true for our setting, in general the regret bounds provided for bounded-memory adversaries are either with respect to a fixed action or a policy class containing policies that do no switch often (Arora et al., 2012; Anava et al., 2015). Our problem formulation for the contextual setup is reminiscent of the recent line of works on contextual MDPs (Azizzadenesheli et al., 2016; Krishnamurthy et al., 2016; Hallak et al., 2015; Modi & Tewari, 2020). However these works either assume that the context vector is fixed for a given episode (allowing for easier planning), or make strong realizability assumptions on the optimal $Q$-function. Finally, our congested bandits formulation of the routing problem is a natural extension of classical work on the online shortest path problem Kalai & Vempala (2005); Awerbuch & Kleinberg (2008); Dani et al. (2007).

## 2 CONGESTED MULTI-ARMED BANDIT

In this section, we model the congestion problem in a multi-armed bandit (MAB) framework. Our setup for congested multi-armed bandits models the congestion phenomenon by allowing the rewards of an arm to depend on the number of times it was played in the past.

Let us consider a MAB setup with $K$ arms, where each arm may represent a possible route. Let us denote by $\Delta$ the size of the window which affects the reward at the current time step. For any time $t$, let $h_t \in \mathcal{H}_\Delta := [K]^\Delta$ denote the history of the actions taken by an algorithm[1] in the past $\Delta$ time steps, that is, $h_t = [a_{t-\Delta}, \ldots, a_{t-1}]$ where $a_\tau$ is the action chosen by the algorithm at time $\tau$. In order to model congestion arising from repeated plays of a single arm, we consider a function $f_{\text{cong}} : [K] \times [\Delta]_+ \to (0, 1)$. This congestion function takes in two arguments: an arm $a$ and the number of times this arm was played in the past $\Delta$ time steps, and outputs a value indicating the *decay* in the reward of arm $a$ arising from congestion.

**Protocol for congestion in bandits.** We consider the following online learning protocol for a learner in our congested MAB framework: At each round $t$, the learner picks an arm $a_t \in [K]$ and observes reward

$$\tilde{r}(h_t, a_t) = f_{\text{cong}}(a_t, \#(h_t, a_t)) \cdot \mu_{a_t} + \epsilon_t \quad \text{where} \quad \epsilon_t \sim \mathcal{N}(0, 1)).$$

Finally, the history changes to $h_{t+1} = [h_{t, 2:\Delta}, a_t]$ where we have used the notation $h_{t, i:j}$ to denote the vector $[h_t(i), \ldots, h_t(j)]$ and $\#(h, a)$ to denote the number of times the action $a$ was played in history $h$.

Each arm $a$ is associated with a mean reward vector $\mu_a$ and the congestion function multiplicatively decreases the reward of that arm. We assume that the learner does not know the exact form of the function $f_{\text{cong}}$ as well as the the mean vector $\mu = \{\mu_a\}_{[K]}$. The objective of the learner is to select the actions $a_t$ which minimizes a notion of policy regret, which we define next.

**Policy regret for congested MAB.** In the standard MAB setup, regret compares the cumulative reward of the algorithm to the benchmark of playing the arm with the highest mean reward at all time steps. However, as described in Section 1, this benchmark is not suitable for our setup. Indeed, the asymptotically optimal algorithm is one which maximizes the average cumulative reward $\rho^* := \max_{\text{alg}} \lim_{T \to \infty} \frac{1}{T} \sum_{t=1}^{T} r(h_t, a_t)$, where the action $a_t$ is the one chosen by the algorithm alg and history $h_t$ is the sequence of actions in the past $\Delta$ time steps.

This asymptotically algorithm corresponds to a stationary policy $\pi^*$ whose selection $a_t$ *only* depends on the history $h_t$. Accordingly, we consider the following class of stationary policies as the comparator for our regret $\Pi = \{\pi : \mathcal{H}_\Delta \to [K]\}$, with size $|\Pi| = K^{K^\Delta}$. Denote by $h_t^\pi$ the history at time $t$ by running policy $\pi$ up to time $t$, we define the policy regret for any algorithm

$$\mathfrak{R}_T(\text{alg}; \Pi, f_{\text{cong}}) := \sup_{\pi \in \Pi} \sum_{t=1}^{T} r(h_t^\pi, \pi(h_t^\pi)) - \sum_{t=1}^{T} \tilde{r}(h_t^{\text{alg}}, a_t) , \tag{1}$$

---

[1] We usually suppress the dependence of this history on the algorithm, but make it explicit whenever it is not clear from context.

---

**Algorithm 1:** CARMAB: Congestion Aware Routing via Multi-Armed Bandits

---

**Input:** Congestion window $\Delta$, confidence parameter $\delta \in (0,1)$, action set $[K]$, time horizon $T$.
**Initialize:** Set $t = 1$
**for** *episodes* $\mathfrak{e} = 1, \ldots, \mathcal{E}$ **do**

> **Initialize episode**
> Set start time of episode $t_{\mathfrak{e}} = t$.
> For all actions $a$ and historical count $j$, set $n_{\mathfrak{e}}(a,j) = 0$ and $N_{\mathfrak{e}}(a,j) = \sum_{s<e} n_s(a,j)$.
> Set empirical reward estimate for each arm and historical count
>
> $$\hat{r}_{\mathfrak{e}}(a,j) = \frac{\sum_{\tau=1}^{t_{\mathfrak{e}}-1} r_\tau \cdot \mathbb{I}[a_\tau = a, j_\tau = j]}{\max\{1, N_{\mathfrak{e}}(a,j)\}}.$$
>
> **Compute optimistic policy**
> Set the feasible rewards
>
> $$\mathcal{R}_{\mathfrak{e}} = \left\{ r \in [0,1]^{A \times (\Delta+1)} \mid \text{for all } (a,j), |r(a,j) - \hat{r}_{\mathfrak{e}}(a,j)| \leq 10 \sqrt{\frac{\log(A\Delta t_{\mathfrak{e}}/\delta)}{\max\{1, N_{\mathfrak{e}}(a,j)\}}} \right\}$$
>
> Find optimistic policy $\tilde{\pi}_{\mathfrak{e}} = \arg\max_{\pi \in \Pi, r \in \mathcal{R}_{\mathfrak{e}}} \rho(\pi, r)$
> **Execute optimistic policy**
> **while** $n_{\mathfrak{e}}(a,j) < \max\{1, N_{\mathfrak{e}}(a,j)\}$ **do**
>
> > Select arm $a_t = \tilde{\pi}_{\mathfrak{e}}(s_t)$, obtain reward $\hat{r}_t$.
> > Update $n_{\mathfrak{e}}(a, \#(s_t, a)) = n_{\mathfrak{e}}(a, \#(s_t, a)) + 1$.

---

where $a_t$ is the action chosen by alg at time $t$ and $r(h_t, a_t) = f_{\mathsf{cong}}(h_t, ) \cdot \mu_{a_t}$. This notion of regret is called policy regret (Arora et al., 2012) because the history sequence observed by the algorithm $\pi^*$ and the algorithm alg can be different from each other – this leads to a situation where choosing the same action $a$ at time $t$ can lead to different rewards for the algorithm and the comparator.

## 2.1 CARMAB: Congested MAB algorithm

We now describe our learning algorithm for the congested MAB problem. At a high level, CARMAB, detailed in Algorithm 1, is based on a reduction of this problem to a reinforcement learning problem with state space $S = \mathcal{H}_\Delta$ and action space $A = [K]$, where the underlying dynamics are known to the learner. With this reduction, CARMAB deploys an epoch-based strategy which plays a optimistic policy $\tilde{\pi}_{\mathfrak{e}}$ computed from optimistic estimates of the reward function.

**Reduction to MDP.** Our congested MAB setup can be viewed as a learning problem in a Markov decision process (MDP) with finite state and action spaces. This MDP $\mathcal{M}_{\mathsf{mab}}$ comprises state space $S = \mathcal{H}_\Delta$ and action space $A = [K]$. The reward function for this MDP is given by $r(s,a) = f_{\mathsf{cong}}(a, \#(s,a)) \cdot \mu_a$ and the deterministic state transitions are given $P(s'|s,a) = \mathbb{I}[s' = [s_{2:\Delta}, a]]$.

**Algorithm details.** Our learning algorithm in this MDP is an upper confidence bound (UCB) style algorithm, adapted from the classical UCRL2 (Jaksch et al., 2010) for learning in finite MDPs. It splits the time horizon $T$ into a total of $\mathcal{E}$ epochs, each of which can be of varying length. In each episode, for every pair $(a,j)$ of action $a$ and historical count $j \in [\Delta]_+$, the algorithm computes the empirical estimate of the rewards $\hat{r}_{\mathfrak{e}}(a,j)$ from observations in the past epoch and maintains a feasible set $\mathcal{R}_{\mathfrak{e}}$ of rewards. This set is constructed such that with high probability, the true reward $r$ belong to this set for each epoch. Given this set, our algorithm computes the optimistic policy

$$\tilde{\pi}_{\mathfrak{e}} = \arg\max_{\pi \in \Pi, r \in \mathcal{R}_{\mathfrak{e}}} \rho_\pi(r) \quad \text{where} \quad \rho_\pi := \lim_{T \to \infty} \frac{1}{T} \sum_{t=1}^{T} r(h_t^\pi, \pi(h_t^\pi)), \tag{2}$$

that is, the policy which achieves the best average expected reward with respect to the optimistic set $\mathcal{R}_{\mathfrak{e}}$. The optimistic policy $\pi_{\mathfrak{e}}$ is then deployed in the congested MAB setup till one of the $(a,j)$ pair doubles in the number of times it is played and this determines the size of any epoch.

**Computing optimistic policy.** In the MDP described above, each deterministic policy $\pi$ follows a cyclical path since the transition dynamics are deterministic and the state space is finite. With

this insight, this problem of finding the optimal policy with highest average reward is equivalent to finding the maximum mean cycle in a weighted directed graph. In our simulations, we use Karp's algorithm (Karp, 1978) for finding these optimistic policy. This algorithm runs in time $O(K^{\Delta+1})$.

## 2.2 REGRET ANALYSIS FOR CARMAB

In this section, we obtain a bound on the policy regret of the proposed algorithm CARMAB. Our overall proof strategy is to first establish that the MDP $\mathcal{M}_{\mathsf{mab}}$ has a low diameter (the time taken to move from one state to another in the MDP), then bounding the regret in each episode $e$ of the process and finally establishing that the total number of episodes $\mathcal{E}$ can be at most logarithmic in the time horizon $T$. Combining these elements, we establish in the following theorem that the regret of CARMAB scales as $\tilde{O}(\sqrt{K\Delta T})$ with high probability.[2]

**Theorem 1** (Regret bound for CARMAB). *For any confidence $\delta \in (0,1)$, congestion window $\Delta > 0$ and time horizon $T > \Delta K$, the policy regret* (1), *of CARMAB is*

$$\mathfrak{R}_T(CARMAB; \Pi, f_{\mathsf{cong}}) \leq c \cdot \Delta^2 K \log\left(\frac{T}{\Delta K}\right) + c\sqrt{K\Delta T \log\left(\frac{\Delta K T}{\delta}\right)} + c\sqrt{T \log\left(\frac{1}{\delta}\right)}$$

*with probability at least $1 - \delta$.*

A few comments on the theorem are in order. Observe that the dominating term in the above regret bound scales as $\tilde{O}(\sqrt{K\Delta T})$ in contrast to the classical regret bounds for MAB which have a $\tilde{O}(\sqrt{KT})$ dependence. This additional factor of $\sqrt{\Delta}$ comes from the fact the stronger notion of policy regret as well as the non-stationary nature of the arm rewards. Additionally, a naïve application of the UCRL2 regret bound to the constructed MDP scales linearly with state space and would correspond to an additional factor of $O(K^{\Delta})$. The CARMAB algorithm is able to avoid this exponential dependence by exploiting the underlying structure in the congested MAB problem. Our regret bound are also minimax optimal – observe that for any constant value of $\Delta$, the lower bounds from the classical MAB setup immediately imply that the regret of any learner should scale as $\Omega(\sqrt{KT})$, which matches the upper bound in Theorem 1.

We defer the complete proof of this to Appendix A but provide a high-level sketch of the important arguments.

**MDP $\mathcal{M}_{\mathsf{mab}}$ has bounded diameter.** The diameter $D$ of an MDP $\mathcal{M}$ measures the number of steps it takes to reach a state $s'$ from a state $s$ using an appropriately chosen policy. The diameter is a measure of the connectedness of the underlying MDP and is commonly studied in the literature on reinforcement learning (Puterman, 2014).

**Definition 1** (Diameter of MDP.). *Consider the stochastic process induced by the policy $\pi$ on an MDP $\mathcal{M}$. Let $\tau(s'|s, \mathcal{M}, \pi)$ represent the first time the policy reaches state $s'$ starting from $s$. The diameter $D$ of the MDP is $D := \max_{s \neq s'} \min_\pi \mathbb{E}[\tau(s'|s, \mathcal{M}, \pi)]$.*

Recall from Section 2.1 that the MDP $\mathcal{M}_{\mathsf{mab}}$ has deterministic dynamics and state space given by the set of histories $\mathcal{H}_\Delta$. Proposition 2 in Appendix A establishes that the diameter of this MDP is at most the window size $\Delta$. With this bound on the diameter, we then show in Lemma 4 that the total regret of the algorithm can be decomposed into a sum of regret terms, one for each episode.

$$\mathfrak{R}_T \leq \sup_{\pi \in \Pi} \sum_{\mathfrak{e}=1}^{\mathcal{E}} \mathfrak{r}_{\mathfrak{e}} + c\sqrt{T \log\left(\frac{T}{\delta}\right)} \quad \text{with } \mathfrak{r}_{\mathfrak{e}} := \sum_{a,j} n_{\mathfrak{e}}(a,j)(\rho_\pi - f_{\mathsf{cong}}(a,j)\mu_a), \quad (3)$$

which holds with probability at least $1 - \delta$. We have used $n_{\mathfrak{e}}(a,j)$ to denote the number of times action $a$ was played by the algorithm when it had a count $j$ in the history and $\rho_\pi$ the average reward of policy $\pi$. Our analysis then proceeds to bound the per-episode regret $\mathfrak{r}_{\mathfrak{e}}$.

---

[2]For clarity purposes, through out the paper we denote by $c$ an absolute constant whose value is independent of any problem parameter. We allow this value of $c$ to change from line to line.

**Regret for episode $\mathfrak{e}$.** In episode $e$, the set of feasible rewards $\mathcal{R}_{\mathfrak{e}}$ is chosen to ensure that with high probability, the true reward $r^*(a, j) = f_{\mathsf{cong}}(a, j) \cdot \mu_j$ belongs to this set. Conditioning on this event, we show that the regret in each episode is upper bounded by the window size $\Delta$ and a scaled ratio of the number of times each action-history $(a, j)$ is played, that is,

$$\mathfrak{r}_{\mathfrak{e}} \leq \Delta + c \sqrt{\log\left(\frac{\Delta K t_{\mathfrak{e}}}{\delta}\right) \sum_{a,j} \frac{n_{\mathfrak{e}}(a, j)}{\sqrt{\max(1, N_{\mathfrak{e}}(a, j))}}} \ . \tag{4}$$

where $t_{\mathfrak{e}}$ is the time at which episode $\mathfrak{e}$ starts and $N_{\mathfrak{e}}(a, j) = \sum_{i < e} n_i(a, j)$. Theorem 1 follows from combining the above with a bound on the total number of episodes $\mathcal{E} \leq c \cdot \Delta K \log\left(\frac{T}{\Delta K}\right)$.

## 2.3 ROUTING WITH CONGESTED BANDITS

We now study an extension of the congested MAB setup where the arms correspond to edges on graph $G = (V, E)$ with a pre-defined start state $s_{\mathsf{G}}$ and goal state $t_{\mathsf{G}}$. In this setup, at each round $t$ the learner selects an $s_{\mathsf{G}}$-$t_{\mathsf{G}}$ path $p_t$ on the graph $G$ and receives reward $\tilde{r}(h_t, e_{i,t}) = f_{\mathsf{cong}}(e_{i,t}, \#(h_t, e_{i,t})) \cdot \mu_{e_{i,t}} + \epsilon_t$ for each $e_{i,t}$ on path $p_t$. The history changes to $h_{t+1} = [h_{t,2:\Delta}, p_t]$.

In comparison to the multi-armed bandit protocol, the learner here selects an $s_{\mathsf{G}}$-$t_{\mathsf{G}}$ path on the graph $G$, the history at any time $t$ consists of the entire set of paths $\{p_{t-\Delta}, \ldots, p_{t-1}\}$, and we assume that the congestion function on each edge $f_{\mathsf{cong}}(h, e)$ depends on the number of times this edge has been used in the past $\Delta$ time steps. The following theorem generalizes the result from Theorem 1 and shows that a variant of CARMAB has regret $\tilde{O}(\sqrt{T})$ for the above $s_{\mathsf{G}}$-$t_{\mathsf{G}}$ online problem.

**Theorem 2** (Regret bound for CARMAB-st). *For any confidence $\delta \in (0, 1)$, congestion window $\Delta > 0$ and time horizon $T$, the policy regret, of CARMAB-st is*

$$\mathfrak{R}_T(\textit{CARMAB-st}; \Pi, f_{\mathsf{cong}}) \leq c \cdot \Delta^2 V E \log\left(\frac{VT}{\Delta E}\right) + c\sqrt{VE\Delta T \log\left(\frac{VE\Delta T}{\delta}\right)} + c\sqrt{VT \log\left(\frac{1}{\delta}\right)}$$

*with probability at least $1 - \delta$.*

The proof of the above theorem is detailed in Appendix A.3.

## 3 LINEAR CONTEXTUAL BANDITS WITH CONGESTION

We now consider the contextual version of the congested bandit problem, where the reward function depends on the choice of arm $a$ as well as an underlying context $x \in \mathcal{X}$. While most of our notation stays the same from the multi-armed bandit setup in Section 2, we introduce the modifications required to account for the context vectors $x_t$. We consider the linear contextual bandit problem where the reward function is parameterized as a linear function of a parameter $\theta_*$ and context-action features $\phi(x_t, a_t)$, that is, $r(h_t, a_t, x_t; \theta_*) := \langle \theta_*, \phi(x_t, a_t) \rangle f_{\mathsf{cong}}(a_t, \#(h_t, a_t))$, where we assume that the context-action features $\phi(x_t, a_t) \subset \mathbb{R}^d$ satisfy $\|\phi(x_t, a_t)\|_2 \leq 1$ and the true parameter $\|\theta_*\|_2 \leq 1$. In contrast to the bandit setup, we expand our policy class to be dependent on the context as well with $\Pi_{\mathcal{X}} := \{\pi : \mathcal{H}_\Delta \times \mathcal{X} \mapsto [K]\}$.

In each round of the linear contextual bandits with congestion game, the learner observes context vectors $\{\phi(x_t, a_i)\}_{[K]}$ and selects action $a_t$. The learner then observes reward $\tilde{r}(h_t, x_t, a_t) = r(h_t, a_t, x_t; \theta_*) + \epsilon_t$ and the history changes to $h_{t+1} = [h_{t,2:\Delta}, a_t]$. The objective of the learner in the above contextual bandit game is to output a sequence of actions which are competitive with the best policy $\pi \in \Pi_{\mathcal{X}}$. Formally, the regret of an algorithm alg is defined to be

$$\mathfrak{R}_T(\mathsf{alg}; \Pi_{\mathcal{X}}, f_{\mathsf{cong}}, \theta_*) := \sup_{\pi \in \Pi_{\mathcal{X}}} \sum_{t=1}^T r(h_t^\pi, \pi(h_t^\pi, x_t), x_t; \theta_*) - \sum_{t=1}^T \tilde{r}(h_t^{\mathsf{alg}}, a_t, x_t; \theta_*) \ . \tag{5}$$

In order to provide some intuition about the algorithm, we start with a simple case where all the contexts are known to the learner in advance and later generalize the results to stochastic contexts.

### 3.1 WARM-UP: KNOWN CONTEXTS

In the known context setup, the learner is provided access to a set of contexts $\{x_t\}$ at the start of the online learning game. Algorithm 2 details our proposed algorithm, CARCB, for this setup.

---

**Algorithm 2:** CARCB: Congested linear contextual bandits with known contexts

---

**Input:** Congestion window $\Delta$, congestion function $f_{\text{cong}}$, action set $[K]$, time horizon $T$,
     contexts $\{x_t\}_{t=1}^T$
**Initialize:** Set $t = 1$, $\theta_1 \sim \text{unif}(\mathbb{B}_d)$
**for** *episodes* $\mathfrak{e} = 1, \ldots, \mathcal{E}$ **do**
    |  **Initialize episode**
    |  Set start time of episode $t_{\mathfrak{e}} = t$.
    |  Let the steps in this epoch $I_e = [t_{\mathfrak{e}}, \ldots, t_{\mathfrak{e}} + 2^{\mathfrak{e}}\Delta]$.
    |  Set the episode policy $\tilde{\pi}_{\mathfrak{e}} = \arg\max_{\pi \in \Pi_{\mathcal{X}}} \sum_{t \in I_{\mathfrak{e}} \setminus \{t_{\mathfrak{e}}, \ldots, t_{\mathfrak{e}}+\Delta\}} r(h_t^\pi, \pi(h_t^\pi, x_t), x_t; \theta_{\mathfrak{e}})$
    |  **Execute estimated policy**
    |  **for** $t = t_{\mathfrak{e}}, \ldots, t_{\mathfrak{e}} + 2^{\mathfrak{e}}\Delta$ **do**
    |    |  Select arm $a_t = \tilde{\pi}_{\mathfrak{e}}(h_t, x_t)$ and observe reward $\hat{r}_t$.
    |  Update $\theta_{\mathfrak{e}}$ via OLS update $\theta_{\mathfrak{e}+1} = \arg\min_\theta \sum_\tau (r_\tau - \langle \theta, \phi(x_\tau, a_\tau) \rangle f_{\text{cong}}(a_\tau, \#(h_\tau, a_\tau)))^2$

---

**Algorithm details.** We again divide the total time $T$ into $\mathcal{E}$ episodes, where the length of each episode $\mathfrak{e} = 2^{\mathfrak{e}}\Delta$. Unlike CARMAB, the algorithm does not maintain any optimistic estimate of the reward parameter $\theta_*$ but simply updates it via an ordinary least squares (OLS) procedure and executes the policy $\tilde{\pi}_{\mathfrak{e}}$ which maximizes this estimated reward function. The core idea underlying this algorithm is that as we observe more samples, our estimate $\theta_{\mathfrak{e}}$ converges to $\theta_*$ and our planner is then able to execute the optimal sequence of actions.

**Regret analysis.** To analyze the regret for CARCB, we study the error incurred in estimating the parameter $\theta_{\mathfrak{e}}$ from the reward samples. To do so, we begin by making the following assumption on the minimum eigenvalue of the sample covariance matrix obtained at any time $t_{\mathfrak{e}}$.

**Assumption 1.** *For* $t > cd$ *and for any sequence of actions* $\{a_1, \ldots, a_T\}$*, we have* $\lambda_{\min}\left(\frac{1}{t} \sum_{\tau \leq t} \phi(x_t, a_t)\phi(x_t, a_t)^\top\right) \geq \gamma$ *, for some value* $\gamma > 0$.

Our bound on the regret $\mathfrak{R}_T$ will depend on this minimum eigenvalue $\gamma$. Later when we generalize our setup to the unknown setup, we will show that this assumption holds with high probability for a large class of distributions. The following theorem shows that the regret bound for $CARCB$ scales as $\tilde{O}(\sqrt{dT} + \Delta)$ with high probability.

**Proposition 1** (Regret bound; known contexts). *For any confidence* $\delta \in (0, 1)$*, congestion window* $\Delta > 0$ *and time horizon* $T > cd$*, suppose that the sample covariance* $\Sigma_t$ *satisfies Assumption 1. Then, the policy regret, defined in eq.* (5)*, of CARCB with respect to the set* $\Pi$ *is*

$$\mathfrak{R}_T(CARCB; \Pi_{\mathcal{X}}, f_{\text{cong}}) \leq \frac{c}{\gamma \cdot c_{\text{min}}} \cdot \sqrt{d(T + \Delta) \cdot \log \frac{\log(T)}{\delta}} + \Delta \log(T) + c\sqrt{T \log\left(\frac{K}{\delta}\right)}$$

*with probability at least* $1 - \delta$ *where* $c_{\text{min}} = \min_{a,j} f_{\text{cong}}(a, j)$.

The proof of the above theorem is deferred to Appendix B. At a high level, the proof proceeds in two steps where first it upper bounds the error $\|\theta_{\mathfrak{e}} - \theta_*\|_2$ for every epoch $\mathfrak{e}$ and then uses this to bound the deviation of the policy $\tilde{\pi}_{\mathfrak{e}}$ from the optimal choice of policy $\pi^*$. In the next section, we generalize this result to the unknown context setup.

## 3.2 UNKNOWN STOCHASTIC CONTEXTS

Our stochastic setup assumes that the context vectors $\{\phi(x_t, a_t)\}$ at each time step are sampled i.i.d. from a *known* distribution. We formally state this assumption[3] next.

**Assumption 2** (Stochastic contexts from known distribution). *At each time instance t, the features* $\phi(x_t, a)$ *for every action* $a \in [K]$ *are assumed to be sampled i.i.d. from the Gaussian distribution* $\mathcal{N}(\bar{x}_a, \Sigma_a)$*, such that* $\alpha_l I \preceq \Sigma_a \preceq \alpha_u I$ *and* $\|\bar{x}_a\|_2 \leq 1$.

---

[3]While our results are stated in terms of the multivariate Gaussian distribution, these can be generalized to sub-Gaussian distribution.

For large-scale recommendation systems for traffic routing which interact with million of users daily, the above assumption on known distributions is not restrictive at all. Indeed these systems have a fair understanding of the demographics of the population which interact with it on a daily basis and the real uncertainty is on which person from this population will be using the system at any time.

The algorithm for this setup is similar to the known context scenario where instead of planning with the exact contexts in the optimistic policy computation step, we obtain the episode policy as

$$\tilde{\pi}_{\mathfrak{e}} = \arg\max_{\pi \in \Pi} \mathbb{E}\Big[ \sum_{t \in I_{\mathfrak{e}} \setminus \{t_{\mathfrak{e}}, \dots, t_{\mathfrak{e}} + \Delta\}} r(h_t^{\pi}, \pi(h_t^{\pi}, x_t), x_t; \theta_{\mathfrak{e}}) \Big] \,,$$

where the expectation is taken with respect to the sampling of context. Our regret bound for this modified algorithm depends on the mixing time of the policy set $\Pi_{\mathcal{X}}$ in an appropriately defined Markov chain.

**Definition 2** (Mixing-time of Markov chain). *For an ergodic discrete time Markov chain $M$, let $d$ represent an arbitrary starting state distribution and let $d^*$ denote the stationary distribution. The $\epsilon$-mixing time $\tau_{\mathsf{mix}}(\epsilon)$ is defined as $\tau_{\mathsf{mix}}(\epsilon) = \min\{t : \max_d \|dM^t - d^*\|_{TV} \leq \epsilon\}$.*

The mixing time of the policy set $\Pi_{\mathcal{X}}$ is given by $\tau_{\mathsf{mix}}^* := \max_{\pi \in \Pi_{\mathcal{X}}} \max_h \tau_{\mathsf{mix}, \pi}(h)$. The following theorem establishes the regret bound for the modified CARCB algorithm, showing that not knowing the context can increase the regret by an additive factor of $\tilde{O}(\sqrt{\Delta \tau_{\mathsf{mix}}^* \cdot T})$.

**Theorem 3** (Regret bound; unknown contexts). *For any confidence $\delta \in (0,1)$, congestion window $\Delta > 0$ and time horizon $T$, suppose that the context sampling distributions satisfy Assumption 2. Then, with probability at least $1 - \delta$, the policy regret of CARCB satisfies*

$$\mathfrak{R}_T(CARCB; \Pi_{\mathcal{X}}, f_{\mathsf{cong}}) \leq c \cdot \sqrt{\alpha_u T \log\left(\frac{K}{\delta}\right)} + c \cdot \sqrt{\Delta \tau_{\mathsf{mix}}^* T \log\left(\frac{K \log(T)}{\delta}\right)}$$

$$+ \frac{c\alpha_u}{c_{\mathsf{min}} \alpha_l} \cdot \sqrt{d(T + \Delta) \cdot \log \frac{\log(T)}{\delta}} + \Delta \log(T) \,. \tag{6}$$

A detailed proof of this result is deferred to Appendix B. Observe that the above bound can be seen as a sum of two terms: $\mathfrak{R}_T \lesssim \sqrt{dT} + \sqrt{\Delta \tau_{\mathsf{mix}}^* T}$ . The first term is a standard regret bound in the $d$ dimensional contextual bandit setup. The second term, particular to our setup, arises because of the interaction of the congestion window with the unknown stochastic contexts. In comparison to the bound in Theorem 1, the window size $\Delta$ interacts only additively in the regret bound surprisingly. The reason for this additive deterioration of regret is that the shared parameter $\theta_*$ allows us to use data across time steps in our estimation procedure – thus, in effect, the congestion only slows the estimation by a factor of $c_{\mathsf{min}}$ which shows up due to the dependence on the minimum eigenvalue.

In order to go from the regret bound in the known context case, Proposition 1, we need to address two key technical challenges: 1) bound the deviation of the reward of policy $\tilde{\pi}_{\mathfrak{e}}$ from the policy which plans with the known sampled contexts, and 2) the context vectors selected by the algorithm $\phi(x_t, a_t)$ satisfy the minimum eigenvalue condition in Assumption 1.

**Deviation from known contexts.** One way to get around this difficulty is to reduce the above problem to the multi-armed bandit on from Section 2. This simple reduction would lead to a regret bound which scales with the size of the context space $|\mathcal{X}|$ which is exponentially large in the dimension $d$. Instead of this, we show that the reward obtained by the distribution maximizer $\tilde{\pi}_{\mathfrak{e}}$ are close to those obtained by the sample maximizer via a concentration argument for random walks on the induced Markov chains. The key to our analysis is the construction of this random walk using policy $\tilde{\pi}_{\mathfrak{e}}$ and then using the following concentration bound from .

**Lemma 1** (Theorem 3.1 in Chung et al. (2012)). *Let $\mathcal{M}$ be an ergodic Markov chain with state space $[n]$ and stationary distribution $d^*$. Let $(V_1, \dots, V_t)$ denote a $t$-step random walk on $M$ starting from an initial distribution $d$ on $[n]$. Let $\mu = \mathbb{E}_{V \sim d^*}[f(V)]$ denote the expected reward over the stationary distribution and $X = \sum_i f(V_i)$ denote the sum of function on the random walk. There exists a universal constant $c > 0$ such that*

$$\Pr(|X - \mu t| \geq \delta \mu t) \leq c \|d\|_{d^*} \exp\left(\frac{-\delta^2 \mu t}{72 \tau_{\mathsf{mix}}}\right) \quad \textit{for } 0 \leq \delta < 1 \,, \tag{7}$$

*where the norm* $\|d\|_{d^*}^2 := \sum_i \frac{d_i^2}{d_i^*}$.

This concentration bound is not directly applicable to our setup because of two reasons: 1) the constructed Markov chain $\mathcal{M}_{\tilde{\pi}_e}$ might not be ergodic, and 2) the norm $\|d\|_{d^*}$ might be unbounded in our setup. By using the fact that the diameter of the MDP $\mathcal{M}_{\tilde{\pi}_e}$ is bounded by $\Delta$, we use an intermediate policy in the time steps $\{t_e : t_e + \Delta\}$ in each episode reach a state starting from which the MDP is shown to be ergodic and have bounded norm $\|d\|_{d^*}$. See Appendix B for details.

**Minimum eigenvalue bound.** In order to obtain a regret bound for the stochastic setup, we need to establish that the covariance matrix formed by these context-action features satisfies the minimum eigenvalue assumption. The challenge here is that the the features $\phi(x_t, a_t)$ are not independent across time – they are correlated since the algorithm's choice at time $t$ depends on the history $h_t$ which in turn depends on the past features. In Appendix B we get around this difficulty by decoupling these dependencies and showing that even after this decoupling, the random variables still satisfy a sub-exponential moment inequality.

## 4 EXPERIMENTAL EVALUATION

In this section, we evaluate both our proposed algorithms, CARMAB and CARCB, in the congested bandit framework and exhibit their no-regret properties.

We generate $K$ arms and assign a base reward of $\hat{r}_j \in (0, 1)$ to each $j \in [K]$. We draw a noise parameter $\epsilon_{t,j}$ for every action $j$ and time step $t$. We set $f_{\text{cong}}(a_t, \#(h_t, a_t)) = 1/\#(h_t, a_t)$. We also set the parameter $\Delta$, which controls the length of the history $h_t$ at time $t$. Then, the observed reward is $\tilde{r}(h_t, a_t) = \frac{\hat{r}_{a_t}}{\#(h_t, a_t)} + \epsilon_{t,a_t}$. We set parameter $\delta$ of algorithm CARMAB to $0.1$. In terms of distributions, we draw $\hat{r}_j$ uniformly in $(0, 1)$ and $\epsilon_{t,j}$ from $\mathcal{N}(0, 0.1)$. In Figure 2, we present how the average regret of the algorithm changes as time progresses for $K = 4$ and different values of the window size $\Delta$ during which congestion occurs.

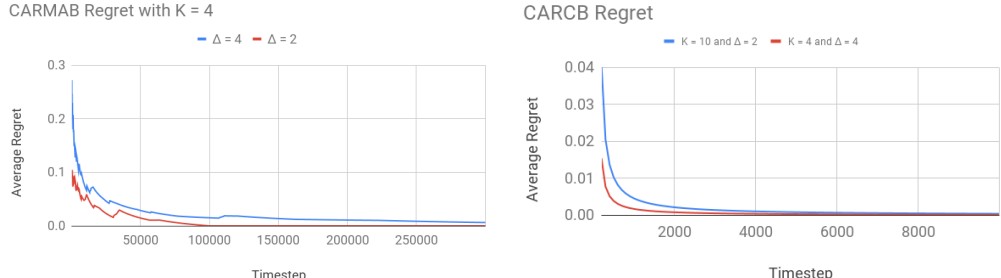

**Figure 2.** (Left) No-regret property of CARMAB. CARMAB is able to learn optimal sequence of arms to play and enjoys a no-regret property. Increasing the size of history window $\Delta$ makes the problem more challenging and requires larger number of time steps. (Right) No-regret property of CARCB.

For evaluating CARCB, again we generate $K$ arms. For each arm $i \in [K]$ and each time step $t \in [T]$, we draw a random context. A context $x_{a,t}$ is a vector of 10 numbers which are drawn uniformly in $(0, 1)$ and then normalized so that the Euclidean norm of the vector is unit. We also draw the true parameter $\theta_*$ in the same way. We assume each arm's context is available to the algorithm at each time step. We use the same noise and congestion function as in the previous section. The observed reward in this setting is:

$$\tilde{r}(h_t, a_t, x_{a,t}, \theta_*) = \frac{x_{a,t}^T \theta_*}{\#(h_t, a_t)} + \epsilon_{t,a_t}$$

In Figure 2 we present how the average regret of CARCB changes over time, in a setting with similar $K$ and $\Delta$ ($K = \Delta = 4$) and a setting with a number of actions larger than the congestion window ($K = 10$ and $\Delta = 2$).

## ETHICS STATEMENT

Our contributions in this work are theoretical in nature and we do not see any ethics related issue arising from this work in the foreseeable future.

## REPRODUCIBILITY STATEMENT

On the theoretical side, we provide detailed proofs for all our theoretical results in the appendix. For the experimental aspect, we have attached our python code as a supplementary file. Furthermore, we have described all necessary hyper parameters and algorithmic details required to reproduce the experiments.

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
