# OpenReview forum: "Congested bandits: Optimal routing via short-term resets"
_ICLR.cc/2022/Conference — ICLR 2022 Submitted_

### Official Review · Reviewer_otHr · 2021-10-31

**Correctness:** 3
**Technical Novelty And Significance:** 2
**Empirical Novelty And Significance:** 2
**Recommendation:** 3
**Confidence:** 3

**Main Review:**

Strengths:

A new bandit model in which arm rewards depend on a past window of actions.

Sharper regret bounds compared to UCLR2 due to the additional structure imposed on arm rewards.

Techniques used to analyze regret under unknown stochastic contexts are sufficiently novel.

Weaknesses:

A niche problem formulation. The current formulation does not truly capture congestion models for traffic routing platforms or other practical routing problems.

The structure of the optimal policy (when mean rewards are known) needs to be better understood.

No lower bounds on regret.

Proposed algorithms are somewhat trivial extensions of UCLR2 and LinUCB.

Contextual regret analysis is performed on multivariate Gaussian distributions only.

Experiments lack adequate comparison, especially comparisons with other MDP-based RL approaches.


**Summary Of The Paper:**

This paper proposes a new structured bandit model, called congested bandits, where an arm’s reward is a decaying function of how many times it has been played during a recent time window. The proposed bandit model aims to address recommendation problems such as route recommendation, in which recommended routes tend to get congested (hence yield lower rewards). Different from prior work on bandits with changing arm reward distributions, the current paper proposes a model in which the effect of congestion resets over Delta time steps. By viewing the problem as structured MDP, the paper develops a variant of UCRL2 that learns to recommend the optimal arm for each congestion state. It is shown that the proposed algorithm achieves a policy regret of tilde{O}(sqrt{KDeltaT}), which significantly improves upon the exponential dependence on the state space of UCRL2. The authors also propose a variant of their algorithm tailored for the linear stochastic contextual bandit setup and analyze the policy regret for this case as well.


**Summary Of The Review:**

This paper proposes a new theoretical bandit model called congested bandits. While there is some novelty in problem formulation, algorithm, and regret analysis, overall, the paper does not adequately address the problem of congestion in real-world routing problems.

In online routing problems, congestion will depend on many other factors such as the exact time of recommendation, speed of the vehicle, etc. One way to relax the current assumption is to assume that the current reward depends on a weighted average of the number of times an arm is played in the past (with decaying weights). This paper proposes a niche theoretical bandit model; however, its practical significance is not clear.

It is not clear how the non-increasing nature of f_cong affects the regret analysis. Does this property have any effect on regret bounds? Is it used as a part of regret analysis to prove sharp bounds? What happens when f_cong is not non-increasing?

The form of the optimal policy when mean rewards are known deserves more discussion. Is the optimal policy an index policy? Is it a threshold-type policy?

Please compare with other methods, like UCLR2 in experiments.

---

> ### Author Response · Authors · 2021-11-17
> **response to reviewer concerns**
>
> We thank the reviewer for their detailed comments. Below, we address the concerns raised in the review.
> * *“The current formulation does not truly capture congestion models for traffic routing platforms or other practical routing problems.”* - The social routing problem is indeed a complex one. Our work is a first step towards incorporating one of the more important aspects of the problem, the notion of congestion, in both the MAB and the contextual MAB setup and our central contribution is to show that indeed learning could be possible with these added congestion constraints.
> * *“The structure of the optimal policy (when mean rewards are known) needs to be better understood.”* - As pointed out in Section 2.1, the optimal time invariant policy with known mean rewards is the maximum mean cycle of an appropriate directed graph, which can be obtained via Karp’s algorithm. If one is interested in a time-varying policy, the solution can then be expressed as the solution to a T-step  dynamic program. Unlike past frameworks where the optimal policy is just to play greedily with respect to known rewards, our formulation does not admit a simple closed form solution to the offline problem.
> * *“No lower bounds on regret.”* - For any constant value of $\Delta$, the lower bounds from the classical MAB setup imply that the regret of any learner should scale as $\Omega(\sqrt{KT})$. This matches the upper bound in Theorem 1. We have added this as a discussion after the statement of the theorem.
> * *“Proposed algorithms are somewhat trivial extensions of **UCLR2** and LinUCB.”* - As pointed out in Section 2.2 (below Theorem 1), running the vanilla UCRL2 algorithm would lead to a regret bound of $O(K^\Delta)$ instead of $O(K\Delta)$ achieved by our proposed algorithm. While the base of our algorithm is derived from UCRL2, we had to make significant changes to incorporate the problem structure and achieve this significantly better regret bound.
> * *“Proposed algorithms are somewhat trivial extensions of UCLR2 and **LinUCB**.”* For the contextual setup, our algorithm is not connected to LinUCB in any form whatsoever. Contrary to any UCB algorithm, we do not maintain any confidence bounds on the parameter space $\Theta$ and our algorithm is actually a very simple iterative ordinary least-squares solution which works in epochs with increasing sizes. The analysis of this simple algorithm involved several technical challenges (highlighted in Section 3.2) and we believe that it is a novel contribution.
> * *“Contextual regret analysis is performed on multivariate Gaussian distributions only.”* - We presented our results for the multivariate Gaussian distribution for ease of presentation but similar results would also hold for contexts sampled from any sub-Gaussian distribution and are much more general. We have added a comment clarifying this in the main paper.

---

### Official Review · Reviewer_w1p9 · 2021-11-01

**Correctness:** 4
**Technical Novelty And Significance:** 2
**Empirical Novelty And Significance:** 1
**Recommendation:** 3
**Confidence:** 4

**Main Review:**

Strengths: This paper raises a new interesting problem, and proposed solid algorithms and derived theoretical analysis on the upper bounds. The results seem to be optimal from my point of view.

However, this paper does not have enough novelty and theoretical depth for getting an accept.

1. The problems of congested bandits in the paper seem trivial for me. For instance, in the congested bandits, there are actually $K\Delta$ numbers or expected rewards we need to learn from the environment. In the classical UCB algorithms, we only need to replace the value $K$ in the confidence intervals to $K\Delta$, and do simple modifications on the pulling strategy, we should be able to get regret upper bounds $\tilde{O}(\sqrt{K\Delta})$ simply. Thus, I do not see enough theoretical novelty from this paper. The other cases have the similar issue.

2. It is better to have results on the lower bounds, without which, it is hard to say the results of this paper are good enough. The authors should explain more why the lower bounds are missing.

**Summary Of The Paper:**

This paper raises and studies a new problem named congested bandits, an extension of the standard multi-armed bandits (MAB) problems. In this setting, the reward of an arm depends on the number of times this arm has been pulled during the past $\Delta$ time. The authors also extended the congested bandits to graph-based congested bandits and congested linear contextual bandits.

The authors proposed new algorithms for the above settings and derived regret upper bounds similar to those of MAB or linear contextual bandits.

To be specific, for congested bandits, the regret upper bound is $\tilde{O}(\sqrt{K\Delta T})$, where $K$ is the number of arms and $T$ is the time horizon. For graph-based congested bandits, the regret upper bound is $\tilde{O}(\sqrt{VE\Delta T})$, where $V$ is the number of nodes and $E$ is the number of edges. For congested linear contextual bandits, the regret upper bound is $\tilde{O}(\sqrt{dT})$.

The authors also conducted numerical simulations to confirm the above theoretical results.


**Summary Of The Review:**

This paper proposed an interesting question, but the algorithms and techniques used in this paper lack enough novelty for being accepted by the conference. Thus, I tend to vote a reject.

---

> ### Author Response · Authors · 2021-11-17
> **response to reviewer concerns**
>
> We thank the reviewer for their comments and address the main concerns raised below.
> * *“The problems of congested bandits in the paper seem trivial for me”* - The proposed solution to solve the congested bandit problem is incorrect on several fronts.
>     1. Reducing the problem to classical bandit with $K\Delta$ arms and using the UCB strategy would require each of the $K\Delta$ arms to be available at all time instances, which is not the case.
>     2. The availability of arms at time step $t$ depends on the actions of the learner in the past $\Delta$ timesteps -- the UCB algorithm is unable to take these dynamics into account while exploring.
>     3. Even if one could run the UCB algorithm (sidestepping the above issues), the regret guarantees would only be with respect to the best arm in hindsight. In contrast, our formulation requires the learner to compete with the best sequence of actions, a notion of dynamic or policy regret, which is not the case in most MAB setups.
>
>     This simplistic view of the problem completely misses the underlying dynamics associated with the problem and the complications that arise because of it. We would respectfully disagree with the assessment that there is no technical novelty in the paper as such simple solutions fail to produce any meaningful result for the proposed setup, which the reviewer themselves find interesting.
> * *“The other cases have the similar issue.”* - Could you please elaborate on the similar issues? We would be happy to address them as well.
> * *“better to have results on the lower bounds”* -- For any constant value of $\Delta$, the lower bounds from the classical MAB setup imply that the regret of any learner should scale as $\Omega(\sqrt{KT})$. This matches the upper bound in Theorem 1. We have added this as a discussion after the statement of the theorem.

---

### Official Review · Reviewer_9u2j · 2021-11-02

**Correctness:** 3
**Technical Novelty And Significance:** 2
**Empirical Novelty And Significance:** 2
**Recommendation:** 3
**Confidence:** 4

**Main Review:**

**1. Motivation**. The congested bandit problem is said to be motivated by the traffic routing problem. However, the proposed model has almost no connection with the classic routing problem. The network routing problem is usually cast as a congestion game [1], in which the cost of an agent depends on the path it chooses and the number of agents choosing the same path. The evolutionary dynamics through which the agents learn to choose the "best" route (a.k.a. the day-to-day equilibrium) is also a well-studied topic. On a specific day, the realized cost of a path still should only depend on the number of agents selecting the path.

[1] Rosenthal, R. W. (1973). A class of games possessing pure-strategy Nash equilibria. International Journal of Game Theory, 2(1), 65-67.

However, the congested bandit model proposed by the authors has a completely different mechanism. Here the reward for selecting a path (arm) depends on the number of times it was played in the past, instead of the total number of agents selecting it at the current iteration. Although it could be understood as another type of "congestion", I see no reason it could be applied to the traffic routing problem.

**2. Model.** It seems that the decision-maker in this bandit is a traffic routing platform. First of all, in such a context, there should be two different types of agents: the platform that recommends the route and the travelers that may (or may not) follow the recommendations. The cost of a path is realized by the travelers rather than the platform. However, the proposed model doesn't have such a bi-level structure; the cost is realized by the platform itself.

Meanwhile, I hope the authors could specify the goal of the platform more clearly. Here are some typical examples in the classic game/bandit literature:

(1). The platform wants to achieve the desired system state. Following Wardrop's first and second principles [2], it could be (i) social optimal, that the total cost of agents is minimized; or (ii) user equilibrium, that no agent has the incentive to change its route. These two models have been widely studied in both operations research and machine learning literature.

[2] Wardrop, J. G. (1952). Road paper. some theoretical aspects of road traffic research. Proceedings of the institution of civil engineers, 1(3), 325-362.

(2). The platform only "cares" about the quality of its recommendation for specific agents. Then the problem may be cast as a **multi-agent bandit** and a "coordinator" aims to identify the best arm [3].

[3] Shi, C., Xu, H., Xiong, W., & Shen, C. (2021). (Almost) Free Incentivized Exploration from Decentralized Learning Agents. arXiv preprint arXiv:2110.14628.

The route recommendation problem may be understood from different perspectives. **Yet, I don't think it can be appropriately modeled without considering these two levels of agents (platform and travelers) independently**. Besides, I also recommend the authors to think about the goal of the platform and the recommendation mechanism (recommend personalized route? would all of the travelers follow the recommendation) more carefully.

**3. Algorithm and regret bound**. It is shown that the congested bandit problem can be reduced to an MDP problem. After this reduction, the overall structure algorithm, as well as the procedure for bounding the regret, could directly follow from the results in [4].  I don't think this paper contributes to this classic problem substantially with new insights in the proof of Theorem 1.

[4] Jaksch, T., Ortner, R., & Auer, P. (2010). Near-optimal Regret Bounds for Reinforcement Learning. Journal of Machine Learning Research, 11(4).

Theorem 2 then provides the regret bound of the proposed algorithm in routing problems. Here the traffic network is defined as a graph $G = (V, E)$. As far as I'm concerned, the regret bound provided in Theorem 2 cannot address how the regret grows clearly. In the graph, the number of nodes, edges and paths (V, E and K) are not totally independent. Specifically, K relies on V and E with a typically expoential relationship. A formula as in Theorem 2 could provide the regret bound of any specific case, yet an overall picture is more desirable. This may be carried out on a graph with a special structure that can we can establish a the relationship between V, E and K explicitly (e.g., a grid).

Minor issue: I don't suggest to "use the V and E to denote both the edge and vertex sets as well as their sizes", which could be extremely confusing. If the authors still want to make this simplification, then this important statement should at least be moved to the main body of the paper.

The congested bandit model is also extended the contextual version. I didn't check the details of the proof in the Appendix for this section. My feeling in general is that it is not that easy to analyze the regret bounds, nevertheless, as contextual bandits and contextual MDP are all well-studied topics, the technical novelty is still not significant.


**Summary Of The Paper:**

This paper studied congested bandits, a special class of multi-armed bandit problems where each arm's reward depends on the number of times it was played in the past few steps. The authors showed the reduction of the congested bandit problem to an MDP problem. A UCRL-type algorithm was then proposed and the regret bound of the algorithm was analyzed.

**Summary Of The Review:**

The overall assumption in the proposed bandit model is that the reward of an arm relies on the number of times it was played in the past. I recommend rejecting this paper in a large part because:
1. This setting cannot address the traffic routing problem which motivated the study.
2. Compared with existing bandit models which take history into consideration, the major difference in this paper is that it considers a short-term reset, which is not a significant modification.
3. The technical contribution is not substantial, as the overall structure of the algorithm and the corresponding regret analysis follows directly from other papers with limited new insights.

---

> ### Author Response · Authors · 2021-11-17
> **response to reviewer concerns**
>
> We thank the reviewer for their comments and discuss some points below.
> * The reviewer is correct that learning in routing has been studied in the context of congestion games. That model specifically focuses on how drivers explore candidate routes over the course of multiple rounds (days). Our approach however is different and does not focus on learning from the players’ perspective. Instead we assume the perspective of maximizing social welfare through the lens of a routing provider. In this regard, each round corresponds to a new vehicle that enters the system and each arm corresponds to a route. That is precisely the reason why arms become worse as they keep getting played in the recent time window: vehicles are being sent on the same route, which gets congested. The popularity of routing platforms hints that our model is well motivated in the regard that drivers seek guidance from a routing provider as opposed to exploring candidate paths over the course of multiple days.
> * Incorporating aspects of incentives in our model is an interesting direction, however we found the base version of the problem to be interesting as well. Even in the context of congestion games, there are of course works that focus on the inefficiency, existence, and computation of equilibria, but there also papers that study welfare maximization such as:
>     1.  Blumrosen, Dobzinski. Welfare maximization in congestion games. ACM EC 2006.
>     2.  Meyers, Schulz. The complexity of welfare maximization in congestion games. Networks 2012.
>
>     Our work focuses on welfare maximization in the presence of unknown costs and in a temporal routing model where traffic requests appear sequentially and traffic departs the network after some time.
> * [Algorithm and regret bound]  As pointed out in Section 2.2 (below Theorem 1), running the vanilla UCRL2 algorithm would lead to a regret bound of $O(K^\Delta)$ instead of $O(K\Delta)$ achieved by our proposed algorithm. While the base of our algorithm is derived from UCRL2, we had to make significant changes to incorporate the problem structure and achieve this significantly better regret bound.
> * For the contextual bandit case, our algorithm is a very simple iterative ordinary least-squares solution which works in epochs with increasing sizes. The analysis of this simple algorithm involved several technical challenges (highlighted in Section 3.2) and we believe that it is a novel contribution. None of the existing techniques and algorithms from the contextual MDP literature actually work for our setup, as we have already discussed in the related work section.
> * [*As far as I'm concerned, the regret bound provided in Theorem 2 cannot address how the regret grows clearly*] There was a typo in the statement of the regret bound and the K in the denominator should actually be an |E| which we have corrected. As you correctly pointed out, the regret bound depends on the structure of the network. Our regret bounds are indeed problem dependent in that sense -- they capture the precise dependencies on the size of the vertex set |V| and the edge set |E|.

---

### Official Review · Reviewer_kNab · 2021-11-03

**Correctness:** 4
**Technical Novelty And Significance:** 3
**Empirical Novelty And Significance:** 3
**Recommendation:** 8
**Confidence:** 3

**Main Review:**

Strengths: The idea in this paper is novel and practical. I believe that the described traffic recommendation is popular in industry, also the theoretical formulation is non-trivial. The content in this paper is substantial. Two bandit settings are well-discussed including almost every possible raised problem. There are still minor places that could be improved, see below.

Weaknesses: (i) there is a constant $c$ in each proposed theorem, I note that the authors claim that the value of $c$ is independent of any parameter and could change from line to line. However, I have no clue how the value of $c$ is scaled in each theorem, which makes the results less informative, and since the regrets highly depends on $c$, more discussions on what $c$ looks like or how to potentially pick $c$ for each theorem is favorable;
(ii) in algorithm 1 CARMAB, in the equation of empirical reward estimate, should $a_t$ and $j_t$ in the numerator be $a_\tau$ and $j_\tau$?
(iii) in the related work section, I would like to know more about differences between existing works and the paper. For example how the work Pike-Burke & Grunewalder (2019) differs from the paper? Current version is not clear to me, like what is instantaneous regret and how it differs from the proposed regret? Is their model formulation the same as the proposed?

**Summary Of The Paper:**

This paper proposes a novel multi-armed bandit framework to formulate the scenario of route recommendation based on dynamic traffic situations. The formulated model fits in both stochastic bandits and linear contextual bandits.

The paper first introduces the stochastic bandit setting with time-varying expected reward for each arm, which depends on the pulling history within a fixed-sized sliding window. The authors propose an algorithm CARMAB, with regret analysis indicating that the regret upper bound scales as $ O(\sqrt{K\Delta T\log(\Delta KT)})$ with high probability.

Then the paper discusses the contextual bandit setting with known contexts. The authors updated the model and propose an algorithm CARCB with regret analysis indicating the regret upper bound scales as $\tilde O(\sqrt{dT+\Delta})$ with high probability. Then the authors further relax the assumption on contexts by allowing unknown stochastic contexts samples from a known distribution. The regret analysis indicates an additional term with respect to  $T$ comparing to the known contexts setting.

**Summary Of The Review:**

I would vote for acceptance of this paper. I believe the idea is novel and practical, also theoretically non-trivial. The content in this paper is substantial. Still there could be some improvable weaknesses, see above.

---

> ### Author Response · Authors · 2021-11-17
> **response to reviewer concerns**
>
> We thank the reviewer for their feedback and are glad that they found the formulation novel. Below, we address the concerns raised in the review.
> * [Constant c for each theorem] Our main results are concerned with how the regret scales as a function of the problem dependent parameters. The constant c in each of the theorems is a problem independent quantity and is not required to be known by the learner in order to implement the algorithm. We have updated the draft to make the constant precise in the proposed algorithms.
> * [Algorithm 1 empirical reward equation] Thanks a lot for pointing out this typo, we have corrected this in the updated version.
> * [Comparison to Pike-Burke & Grunewalder 2019] The model considered in this work is different from the one we study. Their work considers a Bayesian setup where the mean reward function is modelled by a draw from a Gaussian process and depends on the number of rounds since an arm was played. On the other hand, in our work, the reward is a function of the number of times an arm was played in the past $\Delta$ time steps. The notion of instantaneous regret only compares with the class of greedy policies at each timestep as compared to the globally optimal policy. We have added a discussion of this with our problem formulation.

---

> > ### Comment · Reviewer_kNab · 2021-11-23
> > **Response to Authors**
> >
> > Thanks for the response. My questions are mostly addressed. I will keep the score, but I would recommend the authors to further clarify the novelty of their algorithm design and the theoretical analysis. For example, regarding the way the proposed algorithm differs from UCRL2, is it a direct extension or it has some additional subtle designs? Similar suggestion for the regret analysis. On the other hand, I would also recommend the authors to further discuss how their model and algorithm solve practical problems, such as the mentioned traffic routing problems.

---

### Decision · Program_Chairs · 2022-01-20

**Decision:**

Reject

**Comment:**

This paper introduces a new structured bandit problem called congested bandits, where the expected reward for an arm is a decaying function of how many times it has been played recently. This model aims to address problems such as route recommendation, in which recommended routes tend to get congested (hence yield lower rewards). Different from prior work on bandits with non-stationary reward distributions, the effect of congestion in this model resets after Delta time steps. The authors show that this problem can be formulated as a structured MDP and propose a variant of UCRL2 that learns to recommend the optimal arm for each congestion state. They show that the proposed algorithm achieves a policy regret bound that significantly improves upon UCRL2. They also propose a variant of their algorithm tailored for the linear stochastic contextual bandit setup with the associated analysis.

Unfortunately, this is a rather niche problem formulation and it fails to truly capture congestion models for traffic routing platforms (or other practical routing problems) which serve as the main motivation for the paper.  Moreover, the novelty is limited: the setting is very close to existing non-stationary bandit models and the proposed algorithms are straightforward extensions of existing strategies. A possible way for supporting the novelty of the setting could be to improve its theoretical understanding through a lower bounds analysis, which is currently lacking from the paper. Although this paper contains interesting and well-articulated ideas, contributions are not sufficient.